# Design of a Wearable 12-Lead Noncontact Electrocardiogram Monitoring System

**DOI:** 10.3390/s19071509

**Published:** 2019-03-28

**Authors:** Chien-Chin Hsu, Bor-Shing Lin, Ke-Yi He, Bor-Shyh Lin

**Affiliations:** 1Department of Emergency Medicine, Chi Mei Medical Center, Tainan 710, Taiwan; nych2525@gmail.com; 2Department of Biotechnology, Southern Taiwan University of Science and Technology, Tainan 71005, Taiwan; 3Department of Computer Science and Information Engineering, National Taipei University, New Taipei 23741, Taiwan; bslin@mail.ntpu.edu.tw; 4Institute of Imaging and Biomedical Photonics, National Chiao Tung University, Tainan 711, Taiwan; nctukeyiho@gmail.com

**Keywords:** electrocardiogram, conductive gels, noncontact electrode, myocardial ischemia, pacemaker, ventricular premature contraction

## Abstract

A standard 12-lead electrocardiogram (ECG) is an important tool in the diagnosis of heart diseases. Here, Ag/AgCl electrodes with conductive gels are usually used in a 12-lead ECG system to access biopotentials. However, using Ag/AgCl electrodes with conductive gels might be inconvenient in a prehospital setting. In previous studies, several dry electrodes have been developed to improve this issue. However, these dry electrodes have contact with the skin directly, and they might be still unsuitable for patients with wounds. In this study, a wearable 12-lead electrocardiogram monitoring system was proposed to improve the above issue. Here, novel noncontact electrodes were also designed to access biopotentials without contact with the skin directly. Moreover, by using the mechanical design, this system allows the user to easily wear and take off the device and to adjust the locations of the noncontact electrodes. The experimental results showed that the proposed system could exactly provide a good ECG signal quality even while walking and could detect the ECG features of the patients with myocardial ischemia, installation pacemaker, and ventricular premature contraction.

## 1. Introduction

The standard 12-lead electrocardiogram (ECG) is an important tool to assist the diagnosis of myocardial ischemia and arrhythmia. In particular, for the diagnosis of myocardial infarction, 12-lead ECG provides important and meaningful information. The timely treatment of occluded coronary artery is critical to reduce myocardial injury and mortality. In order to save the ischemia myocardium, a 12-lead ECG of the patient must be obtained in emergency or prehospital settings. Myocardial infarction must be identified as early as possible and the patient must be taken to a hospital with a cardiac cath lab. In general, the conventional Ag/AgCl ECG electrodes with conductive gels are used to measure ECG signals. The use of conductive gels can effectively improve the conductivity of the skin electrode interface to acquire a better ECG signal quality [1]. However, the use of conductive gels usually encounters the dying issue of long-time measurement or the dislodgement of electrodes due to wet skin.

In order to improve the above issue of the conventional ECG electrodes, several novel dry electrodes have been proposed in previous studies. Some studies applied the technique of microelectromechanical systems (MEMS) in the development of novel dry electrodes [2]. However, the manufacturing cost and process of these MEMS-based dry electrodes is relatively expensive and complex. Moreover, the measuring method of these MEMS-based dry electrodes are semi-invasive, and this also increases the risk of a skin allergy. Several conductive fabrics, conductive materials, or metals are also used for the development of dry electrodes [3,4,5,6,7,8,9,10,11,12,13,14,15]. In 2013, Zhou et al. proposed microstructure-array metal dry electrodes [16]. In 2012, Jung et al. developed carbon nanotube (CNT)/polydimethylsiloxane (PDMS) composite flexible dry electrodes for ECG measurement [17]. The above dry electrodes are also semi-invasive and could provide a good signal quality in a hairless site. In 2011, Lin et al. proposed a novel foam dry electrode [18,19] to acquire biopotentials without conductive gels. However, the abovementioned dry electrodes have to contact the skin directly and may be unsuitable for measuring biopotentials in a hairy site due to the fact that the hair layer might increase the impedance of the skin-electrode interface.

Different from the above dry electrodes which have to contact with the skin directly, the noncontact dry electrode was developed in recent years. In general, the design of dry electrodes has to minimize the impedance of the skin-electrode interface. Different from other skin-electrode models of other dry electrodes, the skin-electrode interface model of a noncontact dry electrode can be viewed as a coupling capacitance. In 2013, Lin et al. successfully applied the technique of noncontact electrode to acquire a lead I ECG signal [20,21]. Based on our experience of the noncontact electrode design, a wearable 12-lead ECG monitoring system with noncontact electrodes was proposed in this study. By using the properties of the noncontact electrode technique, ECG signal can be measured across thin clothes to avoid contacting a wound of the subject. Moreover, by using the wearable mechanical design, the proposed system can be easily worn and taken off, and the locations of noncontact electrodes can be quickly and easily adjusted. Finally, the performance of the proposed noncontact electrode has also be validated and applied in the detection of myocardial ischemia, installation pacemaker, and ventricular premature contraction.

## 2. Materials and Methods

### 2.1. Fundamental Theory of Noncontact Electrode

The electrode-skin interface models of conventional Ag/AgCl electrode and noncontact electrode are shown in Figure 1a,b. In general, conductive gel has to be applied in the conventional Ag/AgCl electrode to form a conductive layer between the electrode and skin and to reduce the impedance of the electrode-skin interface. Therefore, the equivalent circuit of the conductive gel layer can be simply viewed as a resistor. The skin layer can be viewed as a plate, and its equivalent circuit can be presented as a resistance and a capacitor in parallel. Therefore, the measurement of a biopotential by using the conventional Ag/AgCl electrode has to pass through these equivalent impedances, such as the skin layer, the conductive gel layer, and the metal electrode.

In the electrode-skin interface model of a noncontact electrode, the skin layer and the metal electrode can be viewed as two parallel plates, and the clothes can be viewed as an isolation layer. Therefore, the electrode-skin interface model of a noncontact electrode can be viewed as a capacitor. Figure 2a shows the basic scheme of the designed noncontact electrode. Its major parts contain a metal electrode; an impedance converter, which is used to provide an ultrahigh input impedance to reduce the influence of the variation of the skin-electrode interface impedance; and a high-impedance pathway, which is used to reduce the influence of the bias variation of the unit buffer to ensure it works within the active region. Here, the metal electrode was coated with an isolated layer of solder mask. Figure 2b shows the equivalent electrical model of the used noncontact electrodes. Let V_i_ and V_o_ denote the ECG signal source of the human body and the output of the noncontact electrode, respectively, C_g_ denote the coupling capacitance formed by the skin and the metal electrode plate, R_g_ be the equivalent impedance of the bias pathway, and R_i_ and C_i_ denote the equivalent resistance and the equivalent capacitance of the used operational amplifier, respectively. Therefore, the transfer function of the used noncontact electrode can be expressed as followings.
(1)$$\frac{V_{o}\left(S\right)}{V_{i}\left(S\right)}=\frac{R_{g}//R_{i}//\frac{1}{{\mathrm{SC}}_{i}}}{\frac{1}{{\mathrm{SC}}_{g}}+R_{g}//R_{i}//\frac{1}{{\mathrm{SC}}_{i}}}=\frac{{\mathrm{SC}}_{g}R_{g}R_{i}}{{\mathrm{SC}}_{g}R_{g}R_{i}+{\mathrm{SC}}_{i}R_{g}R_{i}+R_{g}+R_{i}}$$

From the above formula, it showed that decreasing C_i_ could provide the larger amplitude response when C_g_, R_g_, and R_i_ are large enough. In the design of the noncontact electrode, R_i_ and C_i_ are decided by the selection of the used operational amplifier. C_g_ is a coupling capacitor, formed by the metal plate of the noncontact electrode and the skin. In order to increase the value of C_g_, the electrode surface area has to be increased or the distance between the skin and the electrode has to be decreased.

### 2.2. Measurement of Standard 12-Lead Electrocardiogram

The standard 12-lead ECG system is used to collect 12 different ECG signals from different locations simultaneously to completely estimate the vector of electrocardiogram. The measurement of 12-lead ECG system can be simply classified into three parts, including three bipolar limb leads (LeadI, LeadII, and LeadIII), three unipolar limb leads (aVR, aVL, and aVF), and six unipolar chest leads (V1–V6). The measurement of LeadI–LeadIII uses three electrodes placed on the left arm (LA), the right arm (RA), and the left leg (LL) respectively. These unipolar limb leads can be represented as Equations (2)–(4).
(2)$$\mathrm{aVR}=\mathrm{RA}-\frac{1}{2}\left(\mathrm{LA}+\mathrm{LL}\right)=-\frac{1}{2}\left(\mathrm{LeadI}+\mathrm{LeadII}\right)$$
(3)$$\mathrm{aVL}=\mathrm{LA}-\frac{1}{2}\left(\mathrm{RA}+\mathrm{LL}\right)=\mathrm{LeadI}-\frac{1}{2}\mathrm{LeadII}$$
(4)$$\mathrm{aVF}=\mathrm{LL}-\frac{1}{2}\left(\mathrm{RA}+\mathrm{LA}\right)=\mathrm{LeadII}-\frac{1}{2}\mathrm{LeadI}$$

The six unipolar chest leads (V1 to V6) represent the voltage difference between the chest voltage and the average voltages of LA, RA, and LL, and they can be expressed by
(5)$$V_{k}={\mathrm{Chest}}_{k}-\frac{1}{3}\left(\mathrm{RA}+\mathrm{LL}+\mathrm{LA}\right)$$
where $V_{k}$ and ${\mathrm{Chest}}_{k}$ denote the *k*th chest leads and the voltage of the *k*th chest electrode, respectively.

## 3. Design and Implementation of Wearable 12-Lead ECG Monitoring System

The system architecture and photograph of the proposed system is shown in Figure 3, and it mainly contains a wearable mechanical design, a wireless 12-lead ECG acquisition module, and a back-end host system. The wireless 12-lead ECG acquisition module is designed to measure ECG signals and can be embedded into the wearable mechanical design. The wearable mechanical design is designed to be worn easily in daily life and can provide a suitable pressure to avoid the sliding of the wireless 12-lead ECG acquisition module to acquire a good ECG signal quality. Finally, the acquired ECG signal will be transmitted to the back-end host system wirelessly via Bluetooth.

The block diagram of the wireless 12-lead ECG acquisition module is shown in Figure 4, and it mainly consists of several parts: noncontact dry electrodes, multiplexers, a summing amplifier, a front-end amplifier, a microprocessor, and a wireless transmission circuit. First, the biopotentials would be acquired by these noncontact dry electrodes. In order to measure the six unipolar chest leads, the reference signal, combined from the voltages of RA, LL, and LA, has to be obtained. Here, the summing amplifier is used to obtain the combination of the RA, LL, and LA voltages. According to the definition of the 12-lead ECG system, nine dry electrodes would be switched by two multiplexers and, then, would be inputted into the front-end amplifier to obtain the 12-lead ECG signals. The front-end amplifier contains an instrumentation amplifier (AD620, Analog Devices, Norwood, MA, USA; gain = 20), a band-pass filter (gain = 20, and frequency band = 0.1 Hz–150 Hz), and a notch filter of 60 Hz. Then, the preprocessed ECG signals would be digitized by an analog-to-digital converter built into the microprocessor with the sampling rate of 500 Hz and, then, would be sent to the wireless transmission circuit to transmit to the back-end host system. The wearable mechanical design mainly consists of an elastic chest vest and Velcros. By using these Velcros, the designed wireless 12-lead ECG acquisition module could be easily embedded on the elastic chest vest, and it also allows the adjustment of the positions of the noncontact electrodes to reduce the influence of the individual body size difference.

In this system, a commercial tablet was used as the platform for the host system, and a 12-lead ECG monitoring program was also designed to continuously monitor the 12-lead ECG signals. This program would first build the graphical user interface to allow the user operating and setting the program parameters. When clicking the start button, it would call the Bluetooth application programming interface (API) to search the wireless 12-lead ECG acquisition module and to make a connection with this module. After connecting with this module, the thread of DataREC would receive the 12-lead ECG signals, store them into the program buffer, real-time display them on the screen, and store the raw data into the local files.

## 4. Results

### 4.1. Electrical Specifications of Noncontact Electrodes

In this section, the electrical specifications of the designed noncontact electrodes were first investigated. Figure 5a,b shows the magnitude response and phase response of the designed noncontact electrodes. Here, a function generator was used to generate a 1 voltage p-p sine wave with varying frequency (from 0.1 Hz to 1000 Hz), and it was connected to a copper plate coated with an insulation tape as the input of the noncontact electrode. From the experimental results, the magnitude response of the noncontact electrode at the frequency range between 1 Hz and 1000 Hz is stable and flat, and its phase response is also almost linear. Figure 5c shows the referred noise spectrum of the noncontact electrode. In this test, the input of the noncontact electrode was connected to the ground. It showed that the referred noise spectrum of the noncontact electrode in higher frequency would be slightly decayed, and the whole referred noise is almost below $10^{-5}\;V/\mathrm{Hz}$.

Next, the ECG signal quality obtained by the proposed noncontact electrode was compared with that of the conventional ECG electrode with conductive gels. In this experiment, the noncontact electrodes were placed across the chest through a thin T-shirt, and their locations were close to that of these conventional ECG electrodes. Figure 6a–c shows the comparisons between the lead II, aVL, and V6 ECG signals randomly selected from 12-lead ECG signals of different ECG electrodes and their spectra. Here, the ECG machine (PageWriter TC30, Philips, Amsterdam, Netherlands) with Ag/AgCl electrodes in Chi Mei Medical Center, Taiwan was used, and the function of the linear correlation coefficient in the Matlab software was used to estimate the difference between the ECG signal qualities obtained by different ECG electrodes. It showed that the correlation between ECG signals obtained by different electrodes was over 0.95 and that the correlation for the ECG spectra was over 0.99. The ECG signal quality obtained by the proposed noncontact electrodes across a T-shirt was exactly similar to that of the conventional ECG electrodes with conductive gels.

### 4.2. ECG Signal Quality of Wearable 12-Lead ECG Monitoring System under Different Conditions

In this section, the influence of motion artifact on the ECG signal quality of the proposed system was first investigated. Figure 7a,b shows the ECG signals obtained by the proposed system while sitting and walking respectively. While walking, the motion artifact of walking causes the slight baseline swinging of the ECG signals, but its signal quality is still similar to that while sitting. Next, the ECG signals of patients in the emergency room were measured by the proposed system. Figure 8a–c shows the ECGs signal obtained from the patients with myocardial ischemia, installation pacemaker, and ventricular premature contraction, respectively. From the experimental results, the ECG features of Lead III T-wave inversion and the V6 ST-wave depression for myocardial ischemia [22] were measured by the proposed system. In Figure 8b, the pulse waves generated by installation pacemaker [23] in the front of each ECG cycle were also measured. For ventricular premature contraction, its ECG features can be reflected on the broadening QRS wave and the lack of P-wave [24], and the ECG signal in Figure 8c could also present these ECG features of ventricular premature contraction. Next, the effect of the cloth material, the effect of the thickness, and the influence of sweating on the ECG signal quality of the proposed system was also investigated. Figure 9a–d shows the ECG signal measured under different cloth conditions, including materials, thicknesses, and humidity. It shows that the amplitude of the ECG signal is slightly attenuated when the cloth thickness increased. In this experiment, the effect of selecting the cloth material on the ECG signal quality was unobvious. Moreover, the effect of sweating could improve the ECG signal quality.

## 5. Discussion

In Figure 5a–c, in the frequency range between 0.1 Hz and 1000 Hz, the amplitude response of the proposed noncontact electrode is stable, and its phase response is linear. The whole referred noise of the proposed electrode is less than $10^{-5}\;V/\mathrm{Hz}$. From the above electrical specifications, the proposed noncontact electrode is suitable for measuring biopotentials, such as ECG, EEG, etc. In Figure 6a–c, the correlations between the EEG signals and spectra obtained by the proposed noncontact electrode and the conventional electrode with conductive gels are high. The influence of motion artifact on measuring the ECG signal was also investigated. The experimental results showed that the ECG signal quality was also good while walking, due to the fact that the flexibility of the wearable mechanical design might provide a suitable pressure to reduce the shifting of the noncontact electrodes. Moreover, the ECG features of myocardial ischemia, installation pacemaker, and ventricular premature contraction were also measured by the proposed system. Therefore, the reliability and practicability of the proposed system on measuring ECG were good. Moreover, the effect of the cloth condition on the ECG signal quality was also investigated. In Figure 9 a–d, it shows the amplitude of the ECG signal would be attenuated when the cloth thickness increased. Moreover, the effect of sweating could improve the ECG signal quality. This can be explained by the value of Cg in Equation (1) which would increase when the distance between the electrode and skin decreased or the humidity increased, due to the increase in dielectric.

Several dry and noncontact electrodes have been proposed in previous studies, and their specification comparison is summarized in Table 1. In 2013, Zhang et al. proposed a microneedle array (MNA) electrode [25]. The size of this MNA electrode was about $12\times 12{\;\mathrm{mm}}^{2}$. Its substrate and microneedles were made of polydimethylsiloxane (PDMS) and silicon respectively, and finally, the MNA electrode was coated with poly-3,4-ethylenedioxythiophene/polystyrene sulfonate (PEDOT/PPS). This MNA electrode contained the properties of excellent flexibility, good conductivity, and semi-invasive measurement. The MNA electrode could directly penetrate the human stratum corneum to reduce the impedance of the skin-electrode interface and the electrode movement caused from the body friction. However, the manufacturing procedure of the MNA electrode is relatively complex, and its cost is also expensive. Under strenuous exercise, sweating might cause the falling off of the PEDOT/PSS coating, increase the skin-electrode interface impedance, and further affect the signal quality. In 2015, Weder et al. proposed an embroidered electrode [26]. The size of this embroidered electrode was about $70\times 20{\;\mathrm{mm}}^{2}$. Here, polyethylene terephthalate (PET) yarn was used as the electrode substrate, and then, it was coated with silver/titanium (Ag/Ti) to provide a good biocompatibility and good conductivity. However, body hair might easily affect the measurement of this embroidered electrode. In 2011, Liao et al. proposed a spring probe dry electrode [27]. The size of this spring probe dry electrode was about $13\times 13{\;\mathrm{mm}}^{2}$. A 13-mm-diameter copper piece was used as the substrate, and 17 gold-plated spring probes were soldered on the copper piece. The electrode was also coated with silicone. The comb telescopic structure of this electrode allowed it to pass through the hair layer and contact the skin directly. However, the skin-electrode interface impedance might be still higher and affect the signal quality [11]. In 2018, Castro et al. proposed a four-channel contactless capacitively coupled electrocardiography (ccECG) system for the extraction of sleep apnea features [28]. The ccECG system has the advantage of long-term physiological monitoring but the disadvantage of a high variation in the quality of the acquired signals due to its high sensitivity to motion artefacts. This system was only used in sleep apnea; it was not verified and did not guarantee a good performance of the ECG measurement under motion. Different from the above dry and noncontact electrodes, the proposed noncontact electrode could access biopotentials across the clothes without contacting the skin directly. Moreover, a flexible printed circuit board (PCB) was used as the substrate of the proposed electrode. The proposed electrode could easily be embedded into the clothes, and its flexibility could fit the body contour to provide a stable and good signal quality even under motion.

In previous studies, several wearable 12-lead ECG systems have also been designed, and a comparison between the proposed system and other systems is listed in Table 2. In 2016, Boehm et al. proposed a 12-lead ECG T-Shirt [29], and its system size was about $70\times 65{\;\mathrm{mm}}^{2}$. Ten active electrodes were embedded in a T-Shirt to greatly improve the convenience of use. However, it might be difficult to fit the body closely to affect the signal quality under motion. In 2015, Yasunori Tada et al. proposed a 12-lead ECG smart shirt [30], and its system size was about $90\times 28{\;\mathrm{mm}}^{2}$. The system contained 10 dry foam electrodes, and conductive ink lines were also used as the ECG leads. The flexibility of the compressed shirt could help these foam electrodes contact the skin well to provide a good ECG signal quality, even under motion. However, these foam electrodes still have to contact the skin directly to acquire a biopotential, and body hairs might affect their measuring performance. In this study, the size of the proposed system was about $25\times 65{\;\mathrm{mm}}^{2}$, and it contains 9 noncontact electrodes. Different from the above 12-lead ECG system, the flexible PCB was used as the substrate of the proposed noncontact electrodes and a wearable mechanical design was also designed to closely fit the body and provide a suitable pressure to reduce transversal motion and lateral motion. Thus, the proposed system could provide a good ECG signal quality under motion. Moreover, different from other wearable ECG monitoring systems, the proposed systems could access biopotentials across the clothes without contacting with the skin directly.

## 6. Conclusions

In this study, a wearable 12-lead noncontact electrocardiogram monitoring system was proposed and successfully applied in measuring the ECG signals of patients with myocardial ischemia and arrhythmia. From the experimental results, the magnitude and phase responses of the designed noncontact electrodes were suitable for measuring ECG signal, and its referred noise was less than $10^{-5}\;V/\mathrm{Hz}$. The proposed system could provide a good signal quality even while walking. Moreover, the ECG features of myocardial ischemia, installation pacemaker, and ventricular premature contraction could be measured by the proposed system. The properties of the noncontact electrode technique can effectively avoid contacting the wound of the subject. Moreover, the positions of the noncontact electrodes can be easily adjusted to reduce the influence of the individual body size difference. Therefore, the proposed system might be usefully applied in the applications of mobile ECG monitoring in the future.

## Figures and Tables

**Figure 1 sensors-19-01509-f001:**
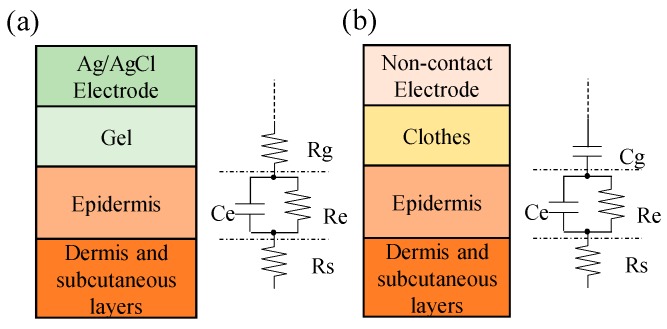
The skin-electrode interface models of (**a**) an Ag–AgCl electrode and (**b**) a noncontact dry electrode.

**Figure 2 sensors-19-01509-f002:**
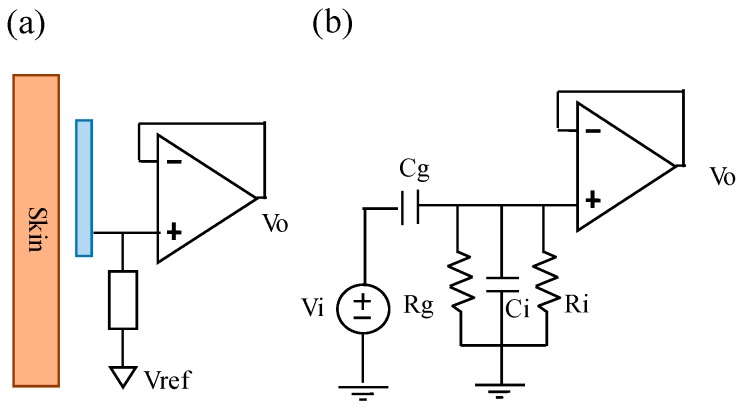
(**a**) A basic scheme and (**b**) the equivalent electrical model of a used noncontact electrode.

**Figure 3 sensors-19-01509-f003:**
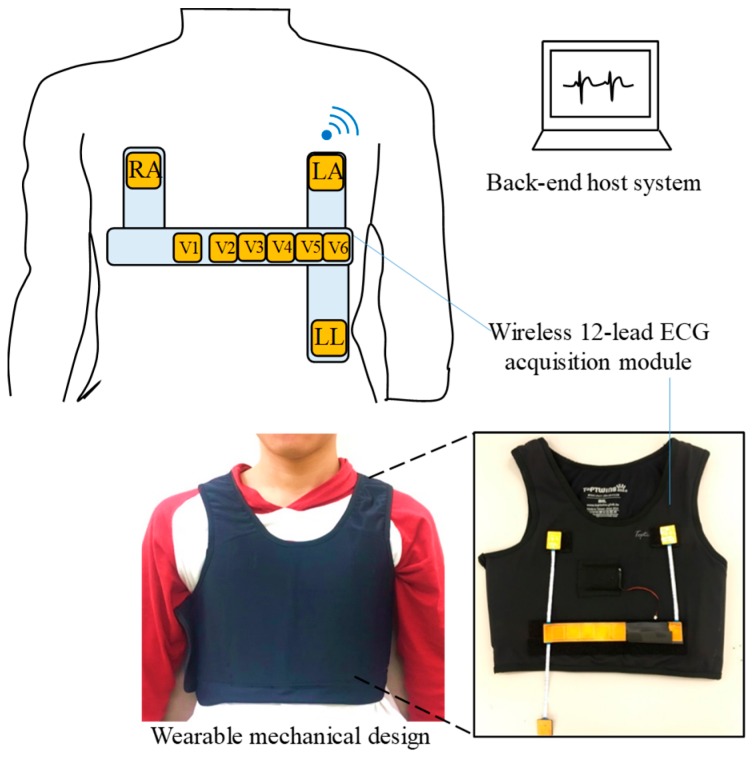
A basic scheme and photograph of the wearable 12-Lead electrocardiogram (ECG) monitoring system.

**Figure 4 sensors-19-01509-f004:**
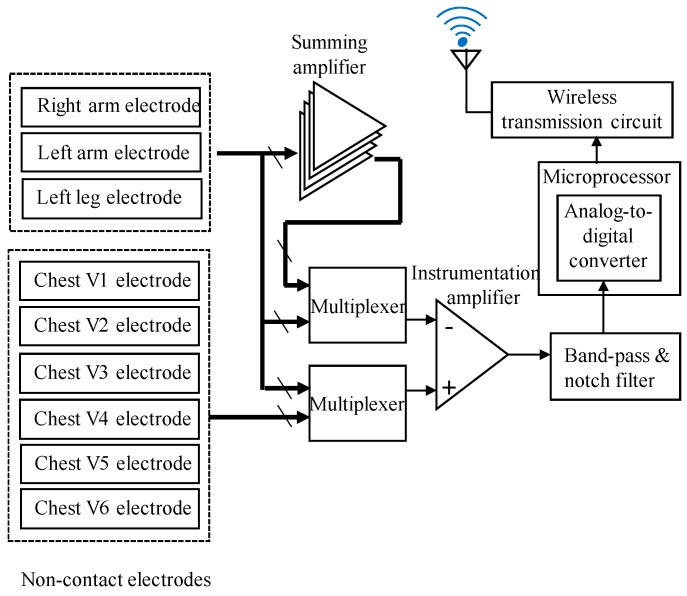
A block diagram of the wearable 12-Lead ECG acquisition module.

**Figure 5 sensors-19-01509-f005:**
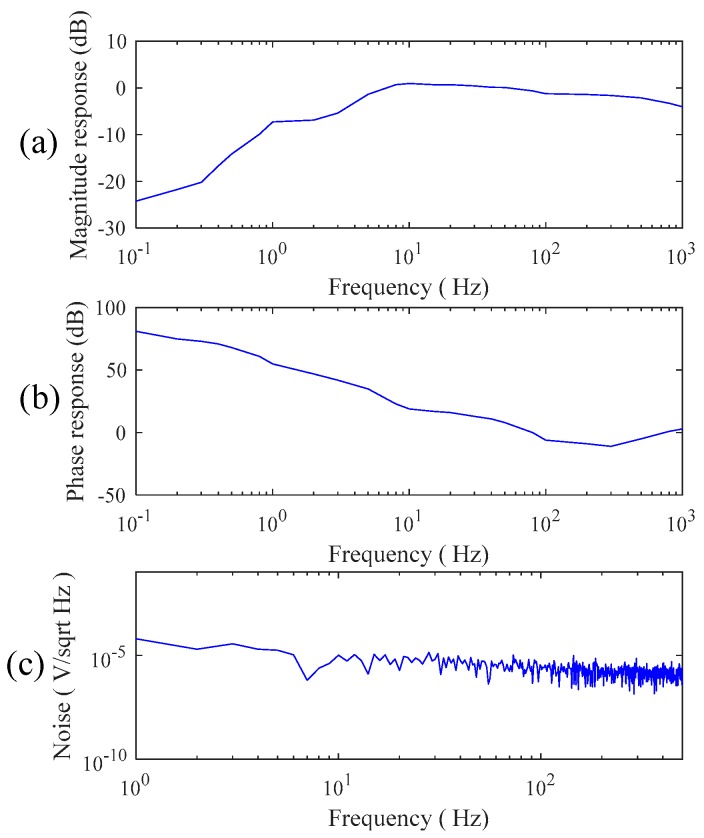
(**a**) The magnitude response, (**b**) phase response, and (**c**) referred noise spectrum of the proposed noncontact electrode.

**Figure 6 sensors-19-01509-f006:**
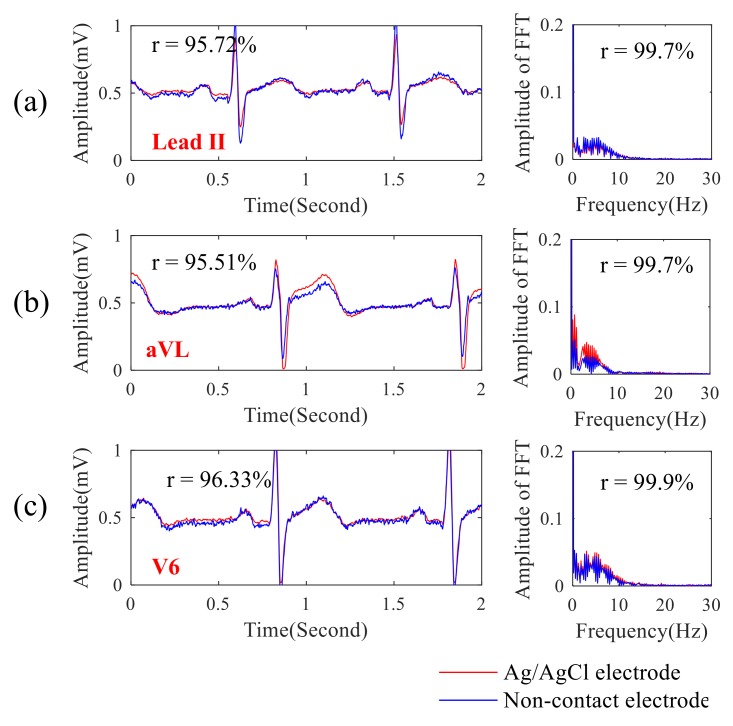
Comparisons between the ECG signals and their spectra obtained by different electrodes: (**a**) lead II, (**b**) aVL, and (**c**) V6.

**Figure 7 sensors-19-01509-f007:**
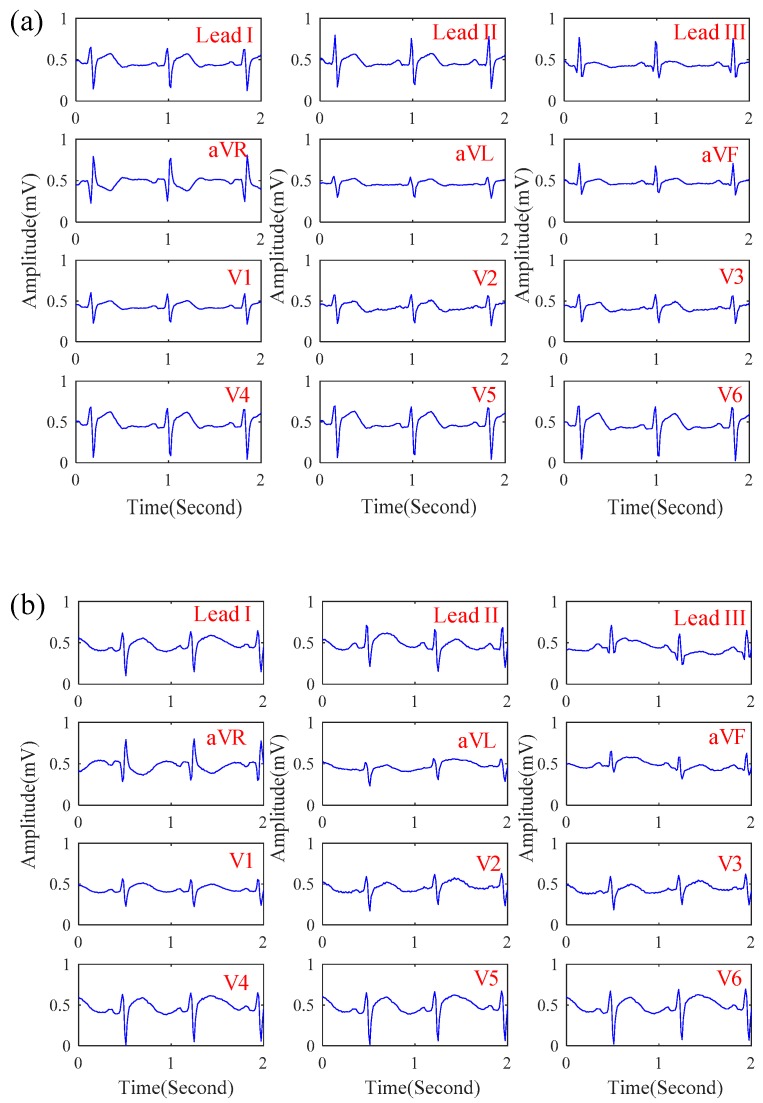
The 12-lead ECG signals obtained by the proposed system when (**a**) sitting and (**b**) walking.

**Figure 8 sensors-19-01509-f008:**
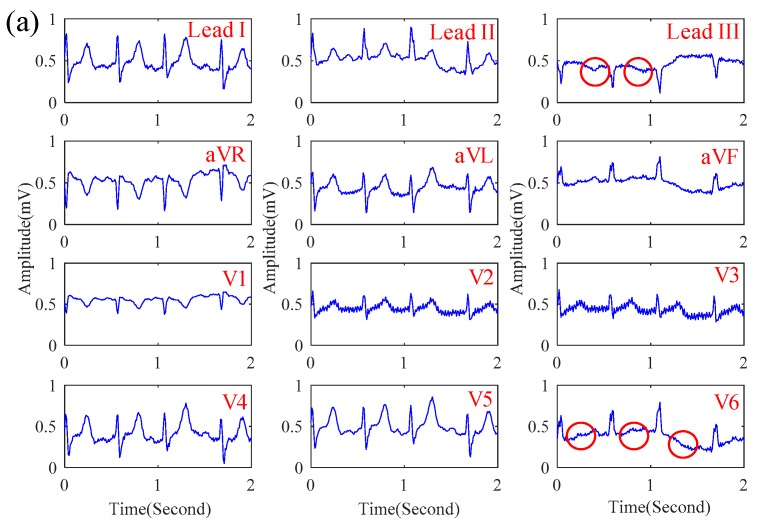
The ECG signal of patients with (**a**) myocardial infarction, (**b**) installation pacemaker, and (**c**) ventricular premature contraction obtained by the proposed system.

**Figure 9 sensors-19-01509-f009:**
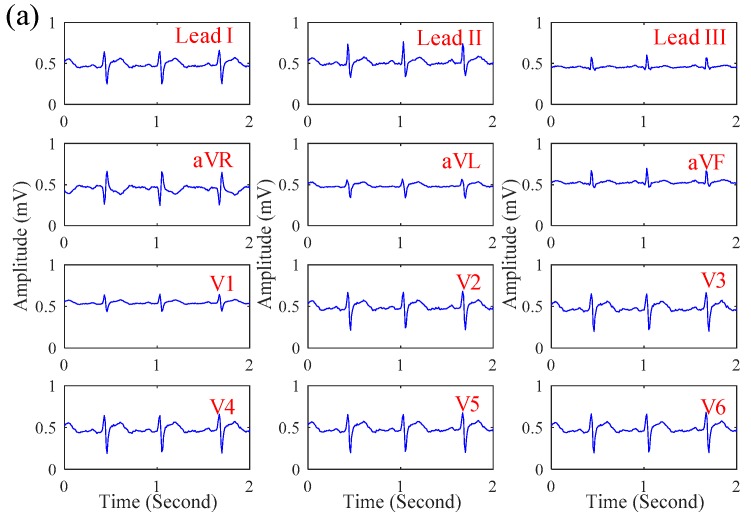
The 12-lead ECG signals obtained by the proposed system under different cloth conditions: (**a**) material: 100% cotton, thickness: 0.8 mm, dry; (**b**) material: 100% cotton, thickness: 1.4 mm, dry; (**c**) material: 80% cotton and 20% polyester, thickness: 0.8 mm, dry; (**d**) material: 100% cotton, thickness: 0.8 mm, sweating.

**Table 1 sensors-19-01509-t001:** The specification comparisons between different dry and noncontact electrodes and the proposed noncontact electrode.

	Zhang et al. [25]	Weder et al. [26]	Liao et al. [27]	Castro et al. [28]	Proposed Electrode
Area of electrode (cm^2^)	1.44	14	1.69	-	6.16
Frequency band (Hz)	0.5–50	-	-	0.5–40	0.1–100
Input-referred noise (V/Hz)	-	-	-	-	6 × 10^−5^
Electrode material	PEDOT/PSS	Ag/Ti	Gold, copper	Ag/AgCl	Copper
Noncontact electrode	No	No	No	Noncontact	Noncontact
Advantages	Excellent flexibility and conductivity, measurement under motion	Good biocompatibility and conductivity	Measurement in hairy site	Good measurement under slight motion (e.g., sleep)	Excellent flexibility, noncontact measurement under motion
Affecting factors	Influence of sweating	Influence of body hairs	Poor skin-electrode interface impedance	Not verified and guaranteed under motion	Thickness of clothing

**Table 2 sensors-19-01509-t002:** The specification comparisons between the proposed 12-Lead ECG monitoring system and other systems.

	Anna Boehm et al. [29]	Yasunori Tada et al. [30]	Proposed System
Operation voltage	3.3 V	-	3.3 V
Amplifier gain	-	-	400 V/V
System of size	70 × 65 ${\mathrm{mm}}^{2}$	90 × 28 ${\mathrm{mm}}^{2}$	25 × 65 ${\mathrm{mm}}^{2}$
Signal resolution	24 bits	-	12 bits
Frequency band (Hz)	2–20 Hz	-	0.1–100 Hz
Wireless transmission	-	XBee	Bluetooth
Power consumption	260 mW	-	150 mW
Advantages	Wearability, long-term monitoring	Wearability, measurement under motion	Wearability, noncontact measurement under motion
Affecting factors	Influence of motion	Influence of body hairs	Thickness of clothing

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
