# Peer review of "Design of a Wearable 12-Lead Noncontact Electrocardiogram Monitoring System"

_sensors, 2019, doi:10.3390/s19071509_

Reviewer 1 Report

This manuscript (ms) describes a 12-lead non-contact wearable electrocardiogram (ECG) monitoring system. The copper electrodes were attached to the vest and ECG signal was collected across a thin T-shirt. The signals were compared with gel electrodes, and were comparable. Also, data were shown for sitting, walking, and some cardiac diseases for verification. The paper is well-written and the results are significant.

Major concerns:

My main major concern is the electrical specification results given in Sec. 4.1. The described procedure states that the signal was connected to the copper plate directly. This nullifies the result as it is no longer a “non-contact” electrode. The authors should revisit this data collection with non-contact setting as their data collection, otherwise these results will be misleading.

Other concerns includes the lack of clarity of electronics (e.g. component model number and values, schematic, precise specification). For instance, Fig. 4 shows the block diagram of the proposed ECG system. It shows 9 electrodes on the left, but it is not clear if all 9 electrodes are connected to both multiplexor or either one multiplexor. Also the results from the Summing amplifier is fed to only one multiplexor, why? What is the circuit used for summing amplifier, considering the ECG signals are too small to use typical op-amp. What is the model of the Inst. Amp? The two multiplexor are shown to connect to two inputs of the Inst. Amp, which is +ve and which is -ve? What are the specifications of the band pass filter? The diagram does not show any utility line noise filter (notch filter). How the utility line interference (which is a major noise and similar in magnitude of ECG signals) is removed? This should be revised and clarified for interested readers.

Another concern is the Fig. 3 where the ECG scheme is shown. It is not clear where are the 9 electrodes. The figure (bottom right) shows 3 electrodes and a strip. Authors should clarify where are the 9 electrodes placed.

During the ECG data collection setup, the authors mention a thin layer of t-shirt separated the skin and the copper electrodes. What are the fiber types? Some artificial cloth fibers are electrically conductive. Have the authors measured electrical characteristics of the cloth? What is the minimum and maximum thickness of t-shirt that the device works if the cloth is non-conductive (e.g. cotton). During the test, did the user sweat (as sweat will create a conductive path from skin to the copper plate, hence no longer non-contact)?

Fig. 6 shows the collected data with the proposed system with Ag/AgCl electrodes. How were the data collected for Ag/AgCl electrodes? A commercial ECG device? Related information is missing.

A minor note is that the authors pitch the device to be suitable for emergency responders as it can be quickly fitted to the patient. But the nature of the vest is that it needs to be tightly in contact with the skin (through a thin t-shirt). Hence it is actually not practical for patients, who might be different body shape and size, and might require minimal handling during transportation. While the results are significant, but it might be wiser for the authors to frame the idea for a different application where this can be utilized.

Some minor concerns are:

1.     Title: Missing “a”.

2.     Pg. 4, line 132: Typo: Frist.

Author Response

Author’s replies to the comments of Editor:

 The authors express their utmost gratitude to the reviewers for their efforts in refereeing the paper and for their constructive suggestions and comments. The paper has been revised taking into account all the reviewers’ comments. The following are item-by-item replies to the reviewers’ comments, including how the authors modified their paper.

Author’s replies to the comments of reviewer1:

[Comment 1]

This manuscript (ms) describes a 12-lead non-contact wearable electrocardiogram (ECG) monitoring system. The copper electrodes were attached to the vest and ECG signal was collected across a thin T-shirt. The signals were compared with gel electrodes, and were comparable. Also, data were shown for sitting, walking, and some cardiac diseases for verification. The paper is well-written and the results are significant.

Answer:

Many thanks for the reviewer’s comments and encouragement.

[Comment 2]

My main major concern is the electrical specification results given in Sec. 4.1. The described procedure states that the signal was connected to the copper plate directly. This nullifies the result as it is no longer a “non-contact” electrode. The authors should revisit this data collection with non-contact setting as their data collection, otherwise these results will be misleading.

Answer:

Many thanks for the reviewer’s comments. In the non-contact electrode, the part of the metal electrode was coated with an isolated layer of solder mask. Moreover, a function generator was used to generate a 1 voltage p-p sine wave with varying frequency (from 0.1 Hz to 1000 Hz), and it was connected to a copper plate coated with an insulation tape as the input of the non-contact electrode. The above description has been added in the modified manuscript.

[Comment 3]

Other concerns includes the lack of clarity of electronics (e.g. component model number and values, schematic, precise specification). For instance, Fig. 4 shows the block diagram of the proposed ECG system. It shows 9 electrodes on the left, but it is not clear if all 9 electrodes are connected to both multiplexor or either one multiplexor. Also the results from the Summing amplifier is fed to only one multiplexor, why? What is the circuit used for summing amplifier, considering the ECG signals are too small to use typical op-amp. What is the model of the Inst. Amp? The two multiplexor are shown to connect to two inputs of the Inst. Amp, which is +ve and which is -ve? What are the specifications of the band pass filter? The diagram does not show any utility line noise filter (notch filter). How the utility line interference (which is a major noise and similar in magnitude of ECG signals) is removed? This should be revised and clarified for interested readers.

Answer:

Many thanks for the reviewer’s comments. According to the reviewer’s comments, the detail for the connections between electrodes, summing circuits, and multiplexors has been modified in Fig. 4. The model of the used Inst. Amp is AD620 (Analog devices, U.S.), and its +ve and -ve has also labeled in Fig. 4. The filter contains a band-pass filter (gain = 20, frequency band = 0.1 Hz – 150 Hz) and a notch filter of 60 Hz. The above information has been added in the revised manuscript.

[Comment 4]

Another concern is the Fig. 3 where the ECG scheme is shown. It is not clear where are the 9 electrodes. The figure (bottom right) shows 3 electrodes and a strip. Authors should clarify where are the 9 electrodes placed.

Answer:

Many thanks for the reviewer’s comments. The placement of the nine electrodes has been added in Fig. 3.

[Comment 5]

During the ECG data collection setup, the authors mention a thin layer of t-shirt separated the skin and the copper electrodes. What are the fiber types? Some artificial cloth fibers are electrically conductive. Have the authors measured electrical characteristics of the cloth? What is the minimum and maximum thickness of t-shirt that the device works if the cloth is non-conductive (e.g. cotton). During the test, did the user sweat (as sweat will create a conductive path from skin to the copper plate, hence no longer non-contact)?

Answer:

Many thanks for the reviewer’s comments. The experimental results for the cloth condition, including material, thickness, sweating, on the ECG signal quality have been added in Fig. 9 (a)-(d). It showed the amplitude of the ECG signal would be attenuated when the cloth thickness increased. Moreover, the effect of sweating could improve the ECG signal quality. This can be explained by the value of Cg in Eq (1) would increase when the distance between the electrode and skin decreased or the humidity increased, due to the increase in dielectric.

[Comment 6]

Fig. 6 shows the collected data with the proposed system with Ag/AgCl electrodes. How were the data collected for Ag/AgCl electrodes? A commercial ECG device? Related information is missing.

Answer:

Many thanks for the reviewer’s comments. Here, the ECG machine (PageWriter TC30, Philips, Netherlands) with Ag/AgCl electrodes in Chi Mei Medical Center, Taiwan was used. The above information has been added in the modified manuscript.

[Comment 7]

A minor note is that the authors pitch the device to be suitable for emergency responders as it can be quickly fitted to the patient. But the nature of the vest is that it needs to be tightly in contact with the skin (through a thin t-shirt). Hence it is actually not practical for patients, who might be different body shape and size, and might require minimal handling during transportation. While the results are significant, but it might be wiser for the authors to frame the idea for a different application where this can be utilized.

Answer:

Many thanks for the reviewer’s comments. According to the reviewer’s comments, the description for the emergency applications has been removed in the modified manuscript.

[Comment 8]

Some minor concerns are:

1. Title: Missing “a”.

2. Pg. 4, line 132: Typo: Frist.

Answer:

Many thanks for the reviewer’s comments. These sentences have been modified.

Reviewer 2 Report

The paper being submitted addresses the use of capacitive (non-galvanic contact) electrodes to capture ECG signals. This subject is not new and in fact several non-contact electrodes have been proposed in the past, which are not referred here in the description of the state of the art. On the other hand, the authors do not provide any manufacturing or implementation details that could show the novelty of the proposed sensor.
The written English also needs to be carefully revised.

Author Response

Author’s replies to the comments of Editor:

 The authors express their utmost gratitude to the reviewers for their efforts in refereeing the paper and for their constructive suggestions and comments. The paper has been revised taking into account all the reviewers’ comments. The following are item-by-item replies to the reviewers’ comments, including how the authors modified their paper.

Author’s replies to the comments of reviewer2:

[Comment 1]

The paper being submitted addresses the use of capacitive (non-galvanic contact) electrodes to capture ECG signals. This subject is not new and in fact several non-contact electrodes have been proposed in the past, which are not referred here in the description of the state of the art. On the other hand, the authors do not provide any manufacturing or implementation details that could show the novelty of the proposed sensor. The written English also needs to be carefully revised.

Answer:

Many thanks for the reviewer’s comments. In the past studies, several non-contact electrodes or 12-lead ECG monitoring system have been respectively proposed. For example:

For non-contact electrode, in 2018, Castro et al. proposed a four-channel contactless capacitively-coupled electrocardiography (ccECG) system for the extraction of sleep apnea features [28]. The ccECG system has advantage of long-term physiological monitoring, but disadvantage of high variation in the quality of the acquired signals due to its high sensitivity to motion artefacts. This system was only used in sleep apnea, it was not verified and did not guarantee to have good performance of ECG measurement under motion.

For 12-lead ECG monitoring system, In 2016, Boehm et al. proposed a 12-lead ECG T-Shirt [29], and its system size was about 70×65 〖mm〗^2. Ten active electrodes were embedded in a T-Shirt to greatly improve the convenience of use. However, it might be difficult to fit the body closely to affect the signal quality under motion. In 2015, Yasunori Tada et al. proposed a 12-lead ECG smart shirt [30], and its system size was about 90×28 〖mm〗^2. The system contained 10 dry foam electrodes, and conductive ink lines were also used as the ECG leads. The flexibility of the compressed shirt could help these foam electrodes contacting with the skin well to provide a good ECG signal quality, even under motion. But, these foam electrodes still have to contact with the skin directly to acquire bio-potential, and body hairs might affect their measuring performance. In this study, the size of the proposed system was about 25×65 〖mm〗^2, and it contains 9 non-contact electrodes.

However, our proposed system is the first wearable 12-lead non-contact ECG monitoring system that can be used in motion. From experimental results, the proposed system was proved to be able to provide a good ECG signal quality under motion.

Table 1 also shows the comparison between several dry and non-contact electrodes and proposed non-contact electrode. Different from other dry and non-contact electrodes listed in Table 1, the proposed non-contact electrode can access bio-potentials across the clothes, without contacting with the skin directly. Moreover, different from other dry and non-contact electrodes proposed in previous studies [25]-[28], a flexible PCB was used as the substrate of the proposed electrode. The proposed electrode could easily be embedded into the clothes, and its flexibility could fit the body contour to provide a stable and good signal quality even in motion.

Table 2 shows the comparison between the proposed 12-lead ECG monitoring system and other systems. Different from other 12-lead ECG system listed in Table 2, the flexible PCB was used as the substrate of the proposed non-contact electrodes and a wearable mechanical design was also designed to closely fit the body and provide suitable pressure to reduce transversal motion and lateral motion. Thus, the proposed system could provide a good ECG signal quality under motion. Moreover, different from other wearable ECG monitoring systems, the proposed systems could access bio-potentials across the clothes, without contacting with the skin directly.

We have carefully revised the sentences and more clear described the innovation of our proposed system in the manuscript.

Round  2

Reviewer 1 Report

his manuscript (ms) describes a 12-lead non-contact wearable electrocardiogram (ECG) monitoring system. The copper electrodes were attached to the vest and ECG signal was collected across a thin T-shirt. 

My concerns are addressed in the revised manuscript. 

Author Response

Author’s replies to the comments of Editor:

The authors express their utmost gratitude to the reviewers for their efforts in refereeing the paper and for their constructive suggestions and comments. The paper has been revised taking into account all the reviewers’ comments. The following are item-by-item replies to the reviewers’ comments, including how the authors modified their paper.

Author’s replies to the comments of reviewer1:

[Comment 1]

his manuscript (ms) describes a 12-lead non-contact wearable electrocardiogram (ECG) monitoring system. The copper electrodes were attached to the vest and ECG signal was collected across a thin T-shirt.

My concerns are addressed in the revised manuscript.

Answer:

Many thanks for the reviewer’s comments and encouragement.

Reviewer 2 Report

The authors made an effort to improve the paper's quality. Its objectives and the characteristics of the contactless electrodes are now more clear, but a deeper charaterization of the electrodes' electrical characteristics could have been provided. The paper is now almost ready for publication. A new revision of the written English is required - for instance, notice the misuse of the past in the following sentences:

...system was shown in Figure 3,
... module was shown in Figure 4
... summing amplifier was used to obtain
... Figure 5 (c) showed the referred noise
... Figure 6 (a)–(c) showed the comparisons .

Also, please rewrite sentence "These unipolar limb leads can be estimated by followings."

Author Response

Author’s replies to the comments of Editor:

The authors express their utmost gratitude to the reviewers for their efforts in refereeing the paper and for their constructive suggestions and comments. The paper has been revised taking into account all the reviewers’ comments. The following are item-by-item replies to the reviewers’ comments, including how the authors modified their paper.

Author’s replies to the comments of reviewer2:

[Comment 1]

The authors made an effort to improve the paper's quality. Its objectives and the characteristics of the contactless electrodes are now more clear, but a deeper charaterization of the electrodes' electrical characteristics could have been provided. The paper is now almost ready for publication. A new revision of the written English is required - for instance, notice the misuse of the past in the following sentences:

...system was shown in Figure 3,

... module was shown in Figure 4

... summing amplifier was used to obtain

... Figure 5 (c) showed the referred noise

... Figure 6 (a)–(c) showed the comparisons .

Also, please rewrite sentence "These unipolar limb leads can be estimated by followings."

Answer:

Many thanks for the reviewer’s comments and encouragement. According to the reviewer’s comments, these sentences have been modified in the revised manuscript.
